# Lipopolysaccharide from *Porphyromonas gingivalis*, but Not from *Porphyromonas endodontalis*, Induces Macrophage M1 Profile

**DOI:** 10.3390/ijms231710011

**Published:** 2022-09-02

**Authors:** Pablo Veloso, Alejandra Fernández, Jessica Astorga, David González-Quintanilla, Alfredo Castro, Alejandro Escobar, Anilei Hoare, Marcela Hernández

**Affiliations:** 1Laboratory of Periodontal Biology, Faculty of Dentistry, Universidad de Chile, Santiago 8380544, Chile; 2Faculty of Dentistry, Universidad Andres Bello, Santiago 8370133, Chile; 3School of Health Sciences, Dentistry, Universidad Viña del Mar, Viña del Mar 2580022, Chile; 4Cellular and Molecular Biology Laboratory, Institute of Dental Sciences, Faculty of Dentistry, Universidad de Chile, Santiago 8380544, Chile; 5Oral Microbiology Laboratory, Faculty of Dentistry, Universidad de Chile, Santiago 8380544, Chile; 6Department of Pathology and Oral Medicine, Faculty of Dentistry, Universidad de Chile, Santiago 8380544, Chile

**Keywords:** *Porphyromonas gingivalis*, *Porphyromonas endodontalis*, lipopolysaccharide, macrophage polarization, toll-like receptor

## Abstract

Apical Lesions of Endodontic Origin (ALEO) are initiated by polymicrobial endodontic canal infection. *Porphyromonas gingivalis* (*Pg*) and *Porphyromonas endodontalis* (*Pe*) lipopolysaccharides (LPS) can induce a pro-inflammatory macrophage response through their recognition by TLR2 and TLR4. However, polarization responses induced by *Pg* and/or *Pe* LPS in macrophages are not fully understood. We aimed to characterize the polarization profiles of macrophages differentiated from THP-1 cells following *Pg* and/or *Pe* LPS stimulation from reference strain and clinical isolates. A modified LPS purification protocol was implemented and the electrophoretic LPS profiles were characterized. THP-1 human monocytes differentiated to macrophages were stimulated with *Pg* and *Pe* LPS. Polarization profiles were characterized through cell surface markers and secreted cytokines levels after 24 h of stimulation. TLR2 and TLR4 cell surfaces and transcriptional levels were determined after 24 or 2 h of LPS stimulation, respectively. LPS from *Pg* induced a predominant M1 profile in macrophages evidenced by changes in the expression of the surface marker CD64 and pro-inflammatory cytokine profiles, TNF-α, IL-1β, IL-6, and IL-12. *Pe* LPS was unable to induce a significant response. TLR2 and TLR4 expressions were neither modified by *Pg* or *Pe* LPS. *Pg* LPS, but not *Pe* LPS, induced a macrophage M1 Profile.

## 1. Introduction

Apical periodontitis (AP) is a prevalent inflammatory disease caused by dental infection comprising the pulpal tissue of the root canal system following a carious lesion of the tooth, which naturally evolves to form an apical lesion of endodontic origin (ALEO), a hallmark of the disease [1]. AP represents the main cause of tooth loss and has been increasingly linked with low-grade systemic inflammation and, consequently, a higher risk of cardiovascular diseases and diabetes mellitus [2,3,4].

The development and progression of ALEO depends on the dynamic balance between the bacterial consortia, their respective virulence factors, and the host’s immune response [5,6,7]. Although endodontic infections have a polymicrobial nature, black-pigmented bacteria, such as *Prevotella* and *Porphyromonas* spp., have been recognized as central agents in the development and progression of the lesion [8,9,10]. Among them, *Porphyromonas endodontalis* (*Pe*) and *Porphyromonas gingivalis* (*Pg*), stand out because of their high prevalence and virulence factors in AP [3,11,12,13,14,15,16,17].

Lipopolysaccharide (LPS), a key virulence factor of Gram-negative bacteria, is a complex glycolipid from the outer membrane, composed of the following three covalently linked domains: lipid A or endotoxin, which is inserted in the outer membrane, the oligosaccharides core and the O-antigen (OAg), which is the outermost and most variable part of the molecule [18]. LPS from *P. gingivalis* has been widely reported to activate inflammatory host response and bone loss during periodontal inflammation [19,20,21]. Likewise, LPS from *P. endodontalis *reduced osteoblast viability, induced the expression of TNF-α and IL-6 via NF-κB, and stimulated osteoclastogenesis and bone resorption in the calvaria mouse model [20,22,23,24,25].

Despite the infectious etiology of AP, the nature, scope, and duration of the immune response against endodontic infection are the main determinants of apical tissue destruction [7,26,27,28]. Resident macrophages are often the first immune cell to recognize pathogen-associated molecular patterns (PAMPs), such as LPS released from Gram-negative bacteria, which is specifically recognized by Toll-like receptors (TLRs) [23]. Macrophages show broad plasticity and differentiation dynamics towards two main phenotypic profiles, pro-inflammatory M1 and anti-inflammatory/healing M2 [28]. M1 macrophages differentiate from the classical activation pathway triggered mainly by the IFN-γ/Jak–STAT1 pathway under the interplay with TLRs [29,30], playing an important role in fighting bacteria and intracellular pathogens, while M2 macrophages result from exposure to IL-4/IL-13 or IL-10 regulated by STAT6 [31]. Symptomatic AP has been associated with M1 over M2 predominance [32] and higher expression of TLR2 in association with its promoter’s hypomethylation [33,34].

Overall, since *P. endodontalis *and *P. gingivalis* coexist with high frequency in AP [35] it is important to decipher the individual and combined effects of *P. endodontalis *and *P. gingivalis* LPS over macrophage differentiation. This study aimed to characterize the polarization profiles and TLR2/4 expression of macrophages differentiated from THP-1 cells following *P. gingivalis* and/or *P. endodontalis *LPS stimulation from reference strain and clinical isolates.

## 2. Results

We characterized and visualized in silver-stained Tris-Tricine gels the electrophoretic profiles of bacterial lysates and their respective LPS extracts from reference strains *P. gingivalis* ATCC 33277, *P. endodontalis *ATCC 35406, and clinical isolates *P. gingivalis* (AC1*Pg*) and *P. endodontalis *(AC2*Pe*). The clinical isolates were previously obtained from patients with asymptomatic AP (AAP).

First, LPS from *P. gingivalis* ATCC 33277 and AC1*Pg* lysates consisted of an unsubstituted Lipid A-core region and a preferential length of lipid A-core substituted with more than eight repeating units of OAg. On the other hand, the electrophoretic profiles of the LPS obtained from both the reference strain and its respective clinical isolate (AC2*Pe*), showed a weaker unsubstituted lipid A-core region compared to that of *P. gingivalis*, and a preferential OAg substitution length of 8 to 14 repeated units (Figure 1a).

LPS purification with the modified TRIzol extraction protocol, consisting of an additional sonication step to ensure wall rupture, and treatments with DNase I, RNase, and proteinase K effectively removed nucleic acids and protein contaminants. The resultant *P. gingivalis* and *P. endodontalis *LPS retained their endotoxic activity (>85 EU/mL) and conserved their structural KDO, a component of the LPS core. The purified *P. gingivalis* LPS obtained from both the reference strain and its clinical isolate (AC1*Pg*), presented an electrophoretic profile similar to that observed in bacterial lysates, showing that the extraction protocol did not affect LPS integrity. In the case of the purified *P. endodontalis *LPS, the unsubstituted lipid A-core region was not visible (Figure 1b). However, both lysates and purified LPS exhibited a preferential length of 8 to 14 OAg repeated units.

Subsequently, we assessed *P. gingivalis* and *P. endodontalis *LPS capacity to induce polarization on THP-1 macrophages. For this purpose, cell cultures were stimulated for 24 h with LPS (10 µg/mL) from *P. gingivalis* ATCC 33277, *P. gingivalis* AC1*Pg*, *P. endodontalis *ATCC 35406, or *P. endodontalis *AC2*Pe* or co-stimulated with *P. gingivalis* ATCC 33277 LPS plus *P. endodontalis *ATCC 35406 LPS (10 μg/mL each). *E. coli* LPS stimulation (10 ng/mL) was used as a positive control.

Macrophage polarization was assessed as the percentage of double-positive cells for M1 macrophages (CD64^+^CD80^+^), or M2 (CD163^+^CD206^+^) and the M1/M2 ratio on THP-1 macrophage. The percentages of single-positive cells and the mean fluorescence intensity (MFI) of each surface marker (Figure 2) were also assessed.

*P. gingivalis* LPS, specifically LPS from AC1*Pg*, showed a tendency to increase the M1 population over unstimulated macrophages by 12.7%. *P. gingivalis* LPS from ATCC 33277 and LPS from AC1*Pg* showed a tendency to increase the M1/M2 ratio to 1.5 and 4.7 respectively, although the changes were not significant (*p* > 0.05).

On the other hand, neither varieties of forms of *P. endodontalis *LPS did not induce changes in surface marker percentages, nor did co-stimulus with *P. gingivalis* ATCC 33277 LPS and *P. endodontalis *ATCC 35406 LPS. None of the evaluated LPS conditions influenced the percentage of double-positive M2 macrophages (Figure 2a–c).

Regarding single positive M1 macrophage populations on unstimulated THP-1 macrophages, 71.3% of the total population were CD64^+^, but only 2% were CD80^+^. *E. coli* LPS stimulus induced a decrease in the CD64^+^ population down to 47.3% but increased CD80^+^ macrophages up to 42.4% (Figure 2d,e). *P. gingivalis* LPS, *P. endodontalis *LPS, and co-stimuli did not significantly change the percentage of CD64^+^ nor CD80^+^ cell population.

CD64 surface levels (MFI) were reduced by *P. gingivalis* LPS, specifically AC1*Pg* LPS, below unstimulated control levels (*p* < 0.05). *P. endodontalis *LPS and the co-stimulus did not promote changes in CD64 nor CD80 surface levels (Figure 2h,i).

Evaluation of single positive M2 macrophage populations showed no significant changes in any of the evaluated conditions. *P. gingivalis* and/or *P. endodontalis *LPS did not change CD163 or CD206 surface levels (MFI), although *E. coli* LPS induced a CD163 increase above unstimulated control (Figure 2f,g,j,k).

In summary, neither *P. gingivalis* LPS nor *P. endodontalis* LPS induced changes in the percentage of cell surface macrophage markers in vitro. Surface levels of CD64 (MFI) were significantly reduced by LPS from clinical isolates of *P. gingivalis*.

Next, the cytokine profiles associated with M1 (TNF-α, IL-1β, IL-6, and IL-12) and anti-inflammatory M2 (IL-10 and IL-1RA) were evaluated in the supernatant of the stimulated THP-1 macrophages (Figure 3). *P. gingivalis* LPS stimulated the secretion levels of pro-inflammatory cytokines similarly to *E. coli* LPS, but in a minor amount, compared with the unstimulated control. Specifically, *P. gingivalis* ATCC 33277 LPS induced increases in TNF-α and IL-1β. Moreover, LPS from AC1*Pg* induced increases in TNF-α, IL-1β, IL-6, and IL-12 compared to unstimulated THP-1 macrophages (*p* < 0.05) (Figure 3). In the case of the anti-inflammatory cytokines, LPS from AC1*Pg* induced an increase of IL-10 compared to unstimulated cells (*p* < 0.05).

*P. endodontalis* LPS stimulation and the co-stimulus with *P. gingivalis* ATCC 33277/*P. endodontalis* ATCC 35406 LPS did not induce changes in the secretion of M1 or M2 cytokines levels compared with *P. gingivalis* ATCC 33277 LPS (Figure 3).

To evaluate the capability of *P. gingivalis* and *P. endodontalis *LPS to induce changes in the expression levels of TLR2 and TLR4, we assessed the percentage of positive macrophage populations for TLR2^+^ and TLR4^+^ and the MFI of TLR2 and TLR4 under the conditions previously described and, in addition, we included PAM_3_CSK_4_ stimulation, a TLR2 ligand, as a positive control. *P. gingivalis* and *P. endodontalis *LPS did not show significant differences in the percentages of TLR2^+^ and TLR4^+^ cells nor induced significant changes in TLR2 or TLR4 MFI compared with non-stimulated control (Figure 4). Although, both positive controls induced significant decrease of TLR2^+^ and TLR4^+^ populations, and TLR2 and TLR4 MFI decreased compared to the unstimulated control.

Correlation analysis between the macrophage surface markers, TLR2, TLR4, and the measured secreted cytokines revealed a strong positive correlation between CD64 and TLRs and a strong negative correlation between CD64 and pro-inflammatory cytokines TNF-α, IL-1β, and IL-6 (Table 1).

We finally evaluated the *P*. *gingivalis* and *P*. *endodontalis* LPS effects on TLR2 and TLR4 mRNA expression in THP-1 macrophages, using the DNA demethylating agent decitabine.

We showed that TLR2 or TLR4 mRNA levels were not affected by *P*. *gingivalis* and *P*. *endodontalis* LPS, nor did previous decitabine treatment (Figure 5).

## 3. Discussion

Black-pigmented bacteria, such as *P*. *gingivalis* and *P*. *endodontalis*, are recognized as central agents in the etiopathogenesis and progression of AP, promoting pro-inflammatory and osteolytic responses due to their diversity of virulence factors [9,13,14,15,16,20]. LPS is a distinctive component of the outer membrane of Gram-negative bacteria and a potent inducer of cytokine secretion. In the case of *P*. *gingivalis* LPS, it has been described to induce an immune response by ligation of both TLR2 and TLR4 because of the heterogeneity of its lipid A structure [36].

In the present work, we purified *P*. *gingivalis* and *P*. *endodontalis* LPS from reference strains (ATCC) and clinical isolates and then we evaluated the polarization profiles in macrophages differentiated from THP-1 after stimulation with these LPSs. We observed that *P*. *gingivalis* LPS induced an M1 profile evidenced by reduction of CD64^+^ cells and increased pro-inflammatory cytokines secretion (TNF-α, IL-1β, IL-6, and IL-12), whereas LPS from *P*. *endodontalis* failed to induce a significant response.

The LPS purification was done using the extraction protocol described by Yi and Hackett, which is a fast, convenient, and low-scale extraction method that uses TRIzol as the main reagent [37]. Nevertheless, we found proteins and nucleic acid traces in our LPS extracts. For this reason, we incorporated enzymatic treatments with DNase, RNase, and proteinase K into the bacterial pellet as a previous step [38,39]. This modification allowed us to remove the contaminants, except for DNA/RNA traces visualized in GelRed-stained agarose gel as an electrophoretic band with a molecular weight smaller than 100 base pairs previously undescribed. Bacterial nucleic acids have immunogenic potential by ligating endosomal TLR7 and TLR9, triggering a response similar to TLR2. To eliminate any trace of nucleic acids, we incorporated the mechanical breakdown step by sonication of the bacterial pellet before the enzymatic treatment [38]. With these modifications to the Yi and Hackett protocol, we obtained *P*. *gingivalis* and *P*. *endodontalis* LPSs free of any detectable traces of proteins and nucleic acids. However, adding extra steps to the original extraction process diminished the performance extraction, fluctuating between 50–90% less dry weight LPS than initially obtained. Despite the latter, the presence of endotoxic activity was confirmed, which was greater than 85 EU/mL, well above that reported in commercial *P*. *gingivalis* LPS (~2 EU/mL) [40]. KDO, a structural component of the LPS inner-core, was also detected in the purified *P*. *gingivalis* and *P*. *endodontalis* LPS, showing that the structure of LPS was not affected by the extraction protocol.

Our *P*. *gingivalis* LPS electrophoretic profile exhibited a high mobility region of unsubstituted lipid A-core clearly defined in the gel and a preferential length of ≥8 OAg repeated units, with a molecular weight between 43 and 100 KDa, consistent with previous descriptions in the literature in other *P*. *gingivalis* strains [41,42,43,44]. On the other hand, compared to *P*. *gingivalis* LPS, *P*. *endodontalis* LPS exhibited a faint lipid A-core region not substituted by OAg and a preferential length of 8 to 14 OAg substitutions repeating units.

It has previously been reported that *P*. *endodontalis* LPS electrophoretic profile in Tris-Glycine gels exhibits a preferential chain length greater than 8 OAg repeated units [45]. In this study, we report that in Tris-Tricine gels, the *P*. *endodontalis* LPS exhibited a preferential chain length greater than 8 OAg repeated units but smaller than the preferential length of *E*. *coli* LPS. Although the electrophoretic profiles of *P. endodontalis* LPS previously presented by Park’s group are slightly different from ours, it is due to methodological differences, since Tris-Glycine gel provides higher resolution to bands with lower electrophoretic mobility while Tris-Tricine gel is suited for a higher resolution to bands with greater electrophoretic mobility [46].

During AP, macrophages exhibit a polarization switch towards M1, whereas disease exacerbation during symptomatic AP is associated with a reduced M2 differentiation profile, based on the reduced surface expression of CD163 along with higher IL-6 and IL-23 levels, which supports the role of the macrophages in the severity of apical lesions that can be triggered by endodontic bacterial antigens [32]. It has been previously reported that *P*. *gingivalis* LPS induces M1 cytokine secretion in bone marrow-derived macrophages once polarized to M1 or M2 [47], as well as M1 cytokine secretion in THP-1 differentiated to macrophages while being polarized with IFN-γ or IL-4 [48]. However, despite the above, the capacity of *P*. *gingivalis* and *P*. *endodontalis* LPS to induce macrophage polarization by analyzing their surface markers or cytokine secretion have not yet been described. In the present study, we phenotypically characterized M1 and M2 macrophage subpopulations based on the expression of their cell surface markers CD64 and CD80 for M1, as well as CD163 and CD206 for M2 [32,49,50,51].

Our positive control, *E*. *coli* LPS, induced an M1 macrophage phenotype with lower CD64 and higher CD80 surface levels in THP-1 macrophages. In the same way, *P*. *gingivalis* LPS induced a reduction of CD64 surface levels, although not on the same level. CD64 is an IgG Fc-γ receptor constitutively expressed on monocytes and macrophages and participates in cellular cytotoxicity by triggering phagocytosis, pro-inflammatory cytokine secretion, ROS production, and antigen presentation to T cells. A decrease in CD64 could suggest a decline in the phagocytic capacity of those macrophages [52,53,54]. In contrast, CD80 is expressed at low basal levels and, like CD86, is a co-stimulatory molecule that, combined with MHC/antigen complex, induces activation, proliferation, and differentiation of T cells [55,56]. For that reason, it might be plausible that macrophages differentiated from THP-1 and stimulated by LPS for 24 h decrease their phagocytic capacity but increase their ability to activate, proliferate and differentiate T cells.

Specifically, *P*. *gingivalis* LPS from our clinical isolate (AC1*Pg*) induced a decrease in CD64 surface levels and an increase in TNF-α, IL-1β, IL-6, and IL-12 cytokine secretion; while *P*. *gingivalis* LPS from ATCC 33277 strain only induced increased secretion of TNF-α and IL-1β in a statistically significant manner. The difference in magnitude response between *E*. *coli* LPS and *P*. *gingivalis* LPS is probably because *P*. *gingivalis* LPS is less immunologically active than *E*. *coli* LPS [57,58,59,60], which could be attributed to its lipid A heterogeneity. *P*. *gingivalis* LPS expresses two varieties of lipid A, a penta-acylated variety, capable of activating TLR4 in a similar way to *E*. *coli* LPS, and a tetra-acylated variety that antagonizes TLR4 by binding it but that does not trigger a response [36,57,61,62].

When comparing *P*. *gingivalis* LPS from different sources, the LPS from clinical isolate induced a more significant response than LPS from ATCC 33277 strain. It has been previously described that clinical isolates of *P*. *gingivalis* with low passages are more virulent than reference strains [63,64], presumably because *P*. *gingivalis* gradually loses the properties necessary to survive straight in vivo, in order to adapt to the in vitro culture conditions [65]. Therefore, since *P*. *gingivalis* from our clinical isolate was obtained from the root canal of a patient with AAP, we can attribute the different results of AC1*Pg* LPS compared to LPS of ATCC 33277 as being due to a higher virulence of *P*. *gingivalis* LPS from clinical isolates over the reference strains.

We also evaluated the immune response triggered by *P*. *endodontalis* LPS, and we observed that it did not induce changes in the secretion of the evaluated cytokines nor the surface levels of M1 or M2 markers. It would be plausible that *P*. *endodontalis* LPS possesses a lipid A heterogeneity similar to that described in *P*. *gingivalis*, but capable of inducing even lower immune responses, It cannot be ruled out that differences in the OAg region between *P*. *endodontalis* and *P*. *gingivalis* LPS may play a role in the low immune response induced in THP-1 macrophages [44,66,67].

Since AP is caused by a polymicrobial infection of the root canal system, we also studied the immune response of THP-1 macrophages co-stimulated with *P*. *gingivalis* and *P*. *endodontalis* LPS. Unexpectedly, LPS co-stimulation did not result in an increased immune response, suggesting that *P*. *endodontalis* LPS could attenuate the immune response induced by *P*. *gingivalis* LPS, by competing and blocking TLR4 and TLR2 [22,36,57], However, whether or not *P*. *endodontalis* LPS can attenuate the immune response in macrophages induced by *P*. *gingivalis* LPS or LPS from other Gram-negative bacteria needs further research. Altogether, these data suggest that macrophage polarization towards M1 during AP progression might be associated with the LPS burden of specific pathogens, such as *P gingivalis*.

Finally, we analyzed the TLR2 and TLR4 surface membrane levels on THP-1 cells differentiated into macrophages after stimulation 24 h with LPS. Surprisingly, both positive controls for ligands for TLR2 and TLR4 induced a reduction in TLR2^+^ and TLR4^+^ population in addition to reduced TLR2 and TLR4 surface levels. These results are difficult to interpret. Although increases in TLR2 and TLR4 levels have been described when stimulated with their respective ligands, other studies have reported a decrease. Specifically, an increase in surface expression of TLR2 and TLR4 was observed in THP-1 cells differentiated for 72 h with PMA (Phorbol 12-myristate 13-acetate) 17 nM and stimulated with *P*. *gingivalis* LPS or *E*. *coli* LPS 1 μg/mL for 24 h [68]. Similarly, THP-1 monocytes stimulated with *P*. *gingivalis* LPS or *E*. *coli* LPS 1 μg/mL for 24 h increased their TLR2^+^ or TLR4^+^ population respectively [69]. On the other hand, a decrease in surface expression of both TLR2 and TLR4, but not of TLR2 and TLR4 mRNA, was described after 24 h of stimulation with *P*. *gingivalis* LPS (1 μg/mL), PAM_3_CSK_4_ (1 μg/mL) or *E*. *coli* LPS (0.1 μg/mL) in human gingival fibroblasts primary cultures [70]. This result could suggest that the protein expression of TLR2 and TLR4 may depend on the concentration of LPS, the cell model, and the conditions of differentiation to macrophages (concentration and time of exposition to PMA). Additionally, the reduction of CD64 was directly correlated with TLR2 and TLR4. In this regard, a directly proportional expression between CD64, TLR2, and TLR4 has been described in monocytes differentiated into macrophages from peripheral blood [71] and neutrophils from tuberculosis patients [72], but not in vitro macrophages differentiated to a pro-inflammatory profile. These antecedents suggest that there might be a functional relationship between the two, which would help explain the reduction in TLRs observed on the surface of LPS-stimulated macrophages in our experimental model.

When *E*. *coli* LPS bonds with TLR4, a hexameric complex is formed, (TLR4/MD-2/LPS)_2_, which activates downstream signaling that induces endocytosis of the complex and, subsequently, transcriptional factor activation [73,74,75,76,77]. It has been reported that bone marrow-derived macrophages (BMDM) internalize TLR4 after 30 min of *E*. *coli* LPS stimulation [76]. In THP-1 differentiated to macrophages and stimulated with *P*. *gingivalis* or *P*. *endodontalis* LPS, or treated with decitabine, a demethylating agent, we did not observe changes in gene expression of TLR2 or TLR4. These results suggest that, in our model, DNA methylation would not play a predominant role in the reduction of TLR2 or TLR4 surface levels. On the other hand, it has been reported in THP-1 cells, that PAM_3_CSK_4_ induced both an increase of TLR2 and an increase of soluble TLR2 (sTLR2). The last one would be mediated by metalloproteinase activity and consequent proteolysis of TLR2 on the membrane surface [78], a process that could be stimulated by PMA and LPS [78,79,80]. These data suggest that the reduction in surface levels of both TLR2 and TLR4 by *E*. *coli* LPS and PAM_3_CSK_4_ could be explained, at least in part, by non-transcriptional mechanisms, such as recycling, lower export to membrane surface, or proteolysis of these receptors.

## 4. Materials and Methods

### 4.1. Bacterial Growth Conditions

*P*. *gingivalis* and *P*. *endodontalis* reference strains (ATCC 33277, ATCC 35406, respectively), and clinical isolates (AC1*Pg*, AC2*Pe*, respectively) previously obtained from AP patients, were inoculated on blood agar 5% supplemented with hemin-menadione (5 μg/mL) and cultured in an anaerobic chamber in an atmosphere of 80% N_2_, 10% CO_2_ and 10% H_2_ (Shel Lab, Cornelius, OR, USA) at 37 °C for 4 days. The bacteria were sub-cultured in 50 mL of brain-heart infusion (BHI) broth supplemented with hemin-menadione (5 µg/mL), grown to late exponential phase, centrifuged at 4500× *g* for 20 min, and stored at −80 °C. The clinical isolates *P*. *gingivalis* (AC1*Pg*) and *P*. *endodontalis* (AC2*Pe*) were identified and confirmed by sequencing of 763 bp fragments corresponding to the V3-V6 regions of the 16S RNA gene.

### 4.2. LPS Purification

LPS was purified by TRIzol extraction protocol [37], which was modified to remove nucleic acids and protein contaminants, as follows: The obtained bacterial pellets were sonicated with an ultrasonic processor, then incubated with DNase (0.1 mg/mL) (Sigma-Aldrich, St. Louis, MO, USA) and RNase (0.2 mg/mL) (Thermo Fisher, Waltham, MA, USA) at 37 °C overnight [38], and, subsequently, with Proteinase K (1 mg/mL) at 60 °C for 1.5 h. Then, they were homogenized in 2 mL TRIzol with 400 μL chloroform and centrifuged at 3900× *g* for 10 min at 4 °C. The upper aqueous phase was recovered and lyophilized overnight. The product was dissolved in 0.375 M MgCl_2_ in 95% ethanol, centrifuged at 4700× *g* for 5 min and the precipitate recovered. LPS purity was verified with a Bradford assay (protein detection limit 0.1 mg/mL) and GelRed stained agarose gel (detection limit 0.1 ng). Bacterial lysates were obtained from cell cultures grown to the exponential phase and adjusted to OD_600_ = 2.0 [81,82]. Purified LPS and bacterial lysates were visualized on silver-stained Tris-Tricine polyacrylamide gels (SDS-PAGE) [46,83] and compared with molecular weight standard #P7706S (BioLabs, Ipswich, MA, USA).

LPS endotoxic activity was determined with Pierce LAL Chromogenic Endotoxin Quantitation Kit (Thermo Fisher, Waltham, MA, USA), and 3-deoxy-D-manno-oct-2-ulosonic acid (KDO), a component of the LPS core, was determined by Purpald test (Sigma-Aldrich, St. Louis, MO, USA) [46].

### 4.3. Cell Culture

THP-1 cells (ATCC TIB-202) were cultured in supplemented RPMI 1640 medium (Corning, New York, NY, USA) with 10% fetal bovine serum (FBS) (Gibco, Thermo Fisher, Waltham, MA, USA) and maintained at 5% CO_2_, 37 °C. THP-1 cells were seeded in triplicates at 1 × 10^6^ cells per well in six-well plates at a concentration of 5 × 10^5^ cells/mL and differentiated to macrophages with 10 nM phorbol 12-myristate 13-acetate (PMA) (Abcam, UK, Cambridge, UK) for 24 h in supplemented RPMI medium with 10% FBS. Non-adherent cells were removed, and adherent cells were washed and incubated for 24 h with RPMI medium with 10% FBS.

THP-1 cells differentiated to macrophages were stimulated with LPS from *P*. *gingivalis*, ATCC 33277 or AC1*Pg* (10 μg/mL), LPS from *P*. *endodontalis* ATCC 35406 or AC2*Pe* (10 μg/mL) or the combination of LPS from *P*. *gingivalis* ATCC 33277 (10 μg/mL) plus LPS from *P*. *endodontalis* ATCC 35406 (10 μg/mL) in RPMI medium without FBS. Cells were stimulated for 2 h to measure TLR2 and TLR4 transcriptional levels or 24 h to assess cell surface proteins levels or cytokine secretion levels [84,85]. Unstimulated cells were used as a negative control. LPS from *E*. *coli* O127: B8 (10 ng/mL) (Sigma-Aldrich, St. Louis, MO, USA) or PAM_3_CSK_4_ (10 ng/mL) (InvivoGen, San Diego, CA, USA) were used as positive controls. Also, a group of cells treated with the demethylation agent 5-aza-2’-deoxycytidine (Decitabine, 500 nM; Sigma-Aldrich, St. Louis, MO, USA) was included to explore whether DNA methylation changes influenced *P*. *gingivalis* or *P*. *endodontalis* LPS-induced TLR2 and TLR4 gene expression.

### 4.4. Flow Cytometry

Cells were placed in a V bottom 96-well plate in PBS for complete staining. Macrophages were stained with Fixable Viability Dye eFluor 780 (Invitrogen, Thermo Fisher, Waltham, MA, USA) 1:1000 in PBS for 30 min at 4 °C in the dark to discard the dead cells. Later, the cells were incubated with a mixture of all fluorochrome-conjugated antibodies at 1:25 each in PBS 2% FBS, for 30 min at 4 °C in the dark. The cells were single-stained under the same conditions as controls. After washing with PBS, the cells were fixed with 2% paraformaldehyde (Invitrogen, Thermo Fisher, Waltham, MA, USA) and analyzed by flow cytometry (LSR Fortessa X-20^®^; Becton Dickinson Immunocytometry Systems, Franklin Lakes, NJ, USA). Flow cytometric data analysis was performed with Flow Jo v10 software (TreeStar, Ashland, OR, USA). The following mouse monoclonal anti-human antibodies conjugated with their respective fluorochromes were used: CD64 conjugated with Brilliant Violet 510 (BV510) and CD80 conjugated with fluorescein isothiocyanate (FITC) were used to identify M1 macrophage subpopulations; CD163 conjugated with PE/Cy7 and CD206 conjugated with Brilliant Violet 421 (BV421) were used to identify M2 macrophages subpopulations; TLR2 conjugated with APC and TLR4 conjugated with PE were also used (Biolegend, San Diego, CA, USA). Fixable Viability Dye (FVD) eFluor 780 (Invitrogen, Thermo Fisher, Waltham, MA, USA).

THP-1 macrophages were first gated according to their forward- and side-scatter profiles. Negative FVD cells were gated to discard dead cells. For the determination of the frequencies of M1 and M2 macrophage subpopulations, cells were gated over CD64^+^CD80^+^ or CD163^+^CD206^+^. The threshold in each case was determined by the comparison with unstained samples. The M1/M2 ratio was calculated as the frequency of M1 double-positive macrophages divided by the frequency of the M2 double-positive macrophages in each sample. The intensity of expression of each specific cell surface marker was expressed as MFI.

### 4.5. Cytokine Secretion

Supernatant from stimulated cells was collected and stored at −80 °C. Cytokines quantification was performed using a multiplex bead-based immunoassay (Human Magnetic Luminex Assay^®^, R&D Systems, Minneapolis, MN, USA) for TNF-α, IL-1β, IL-6, IL-12p70, IL-10, and IL-1RA, following the manufacturer instructions. Data from the multiplex panel were read using a dedicated platform (Magpix^®^, Millipore, MO, USA) and software (Milliplex AnalystR^®^, Viagene Tech, Minneapolis, MN, USA), according to the manufacturer instructions.

### 4.6. RNA Expression

TLR2 and TLR4 relative mRNA levels were determined by qRT-PCR, and cDNA was amplified using their appropriate 5′-3′ forward primer and 5′-3′ reverse primer sets, respectively, as follows: CTCAACACGGGAAACCTCAC and CGTCCACCACTAAGAACG for 18S rRNA, CTCTCGGTGTCGGAATGTC and AGGATCAGCAGGAACAGAGC for TLR2 and CCCTCCCCTGTACCCTTC and TCCCTGCCTTGAATACCTTC for TLR4. All primer sets and reagents (KAPA™SYBR^®^ Fast; KAPA Biosystems, Woburn, MA, USA) were used in qPCR equipment (StepOnePlus^®^; Applied Biosystems, Singapore) as follows: 95 °C for 3 min and 40 cycles of 95 °C for 30 s and 60 °C for 30 s. To detect nonspecific product formation and false-positive amplification, a melt curve of 95 °C for 15 s, 60 °C for 1 min, and 95 °C for 15 s was performed. Relative quantification of mRNA was analyzed using the 2^–ΔΔCT^ method. The normalization of gene expression was done against the 18S rRNA.

### 4.7. Data Analysis

The Shapiro-Wilk test was used to determine the distribution of the continuous data. Inferential analyses were performed using the Kruskal-Wallis test for non-normally distributed data. The statistical analysis was performed using STATA 12^®^ (StataCorpLP, College Station, TX, USA). The figures were performed in Flow Jo v10 software (TreeStar, Ashland, OR, USA) and Graph Prism 5 (GraphPad Software, Inc., San Diego, CA, USA).

## 5. Conclusions

In summary, *P*. *gingivalis* LPS induces a macrophage polarization profile predominantly towards M1, similarly, although less evidently, than that generated by *E*. *coli* LPS, based on their cell surface markers and secreted cytokines. On the other hand, *P*. *endodontalis* LPS did not cause a clear immune response in THP-1 macrophages. *P*. *gingivalis* and *P*. *endodontalis* LPS stimulation did not induce TLR2 or TLR4 levels changes in THP-1 macrophages. These data suggest that macrophage polarization towards M1 during AP progression might be associated with the LPS burden of specific pathogens, such as *P gingivalis*.

## Figures and Tables

**Figure 1 ijms-23-10011-f001:**
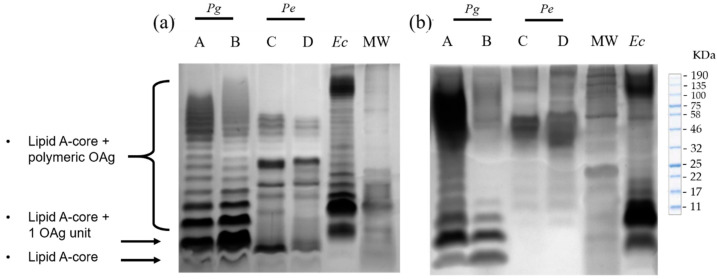
Representative silver-stained Tris-Tricine gels of *P. gingivalis* and *P. endodontalis *LPS; (**a**) Bacterial lysates. (**b**) Purified LPS. A: *P. gingivalis* reference strain ATCC 33277; B: *P. gingivalis* from clinical isolate AC1*Pg*; C: *P. endodontalis *reference strain ATCC 35406; D: *P. endodontalis *from clinical isolate AC2*Pe*; *Ec*: commercial *E. coli* LPS (BioLabs, Ipswich, MA, USA). MW: Molecular Weight Ladder.

**Figure 2 ijms-23-10011-f002:**
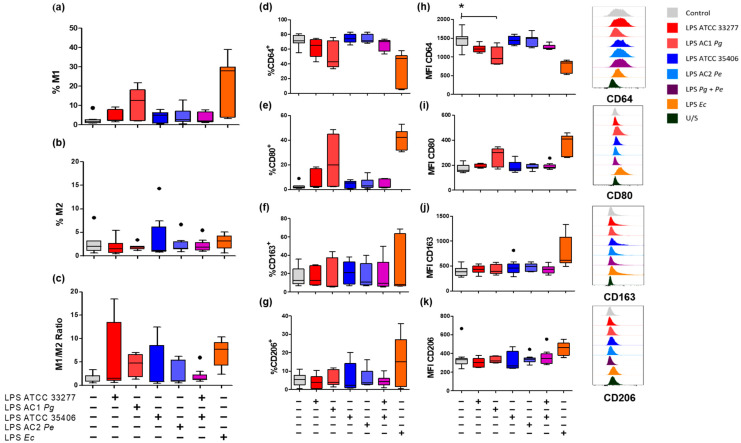
Percentages of M1 and M2 macrophage populations in LPS-stimulated macrophages in vitro. Box and whisker plots represent the median and interquartile range of (**a**) Percentage of CD64^+^CD80^+^ for M1 macrophages. (**b**) CD163^+^CD206^+^ for M2 macrophages. (**c**) M1/M2 ratio. (**d**) CD64^+^, (**e**) CD80^+^ (**f**) CD163^+^ and (**g**) CD206^+^ cell percentage, respectively. (**h**) Mean fluorescence intensity (MFI) of CD64, (**i**) CD80, (**j**) CD163 and (**k**) CD206 respectively of THP-1 macrophages in vitro stimulated for 24 h with LPS. U/S Unstained. Independent experiments were performed in triplicate. Data were analyzed by the Kruskal-Wallis test. (●) Represents outlier data. (*n* = 3) * *p* < 0.05.

**Figure 3 ijms-23-10011-f003:**
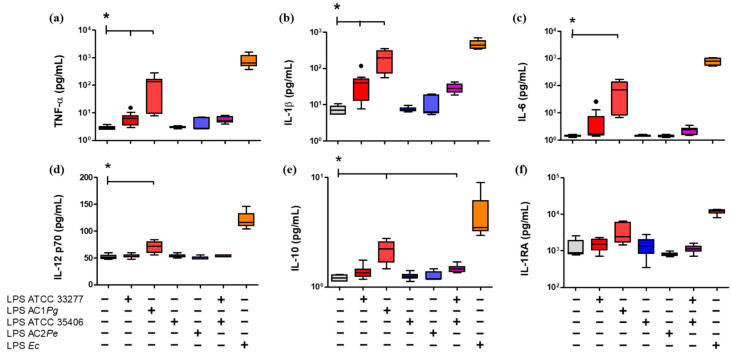
Cytokine secretion LPS-stimulated macrophages in vitro. Box and whisker plots represent the median and interquartile range of pro-inflammatory cytokines (**a**) TNF-α, (**b**) IL-1β, (**c**) IL-6, and (**d**) IL-12p70, and anti-inflammatory cytokines (**e**) IL-10 and (**f**) IL-1RA, detected in the supernatant of THP-1 macrophages in vitro culture stimulated for 24 h with LPS. U/S Unstained. Experiments were performed in triplicate. Data were analyzed by the Kruskal-Wallis test. (●) Represents outlier data. (*n* = 3) * *p* < 0.05.

**Figure 4 ijms-23-10011-f004:**
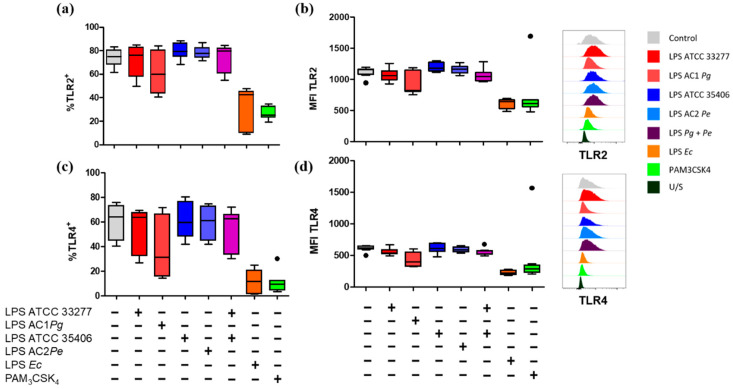
Percentage of single stain and expression levels of TLR2 and TLR4 in LPS or lipopeptide-stimulated macrophages in vitro. Box and whisker plots represent the median and interquartile range of (**a**) TLR2^+^ and (**b**) TLR4^+^ percentage. (**c**) Mean fluorescence intensity (MFI) of TLR2 and (**d**) TLR4 of THP-1 macrophages in vitro stimulated for 24 h with LPS or PAM_3_CSK_4_. U/S Unstained. Experiments were performed in triplicate. Data were analyzed by the Kruskal-Wallis test. (●) Represents outlier data. (*n* = 3).

**Figure 5 ijms-23-10011-f005:**
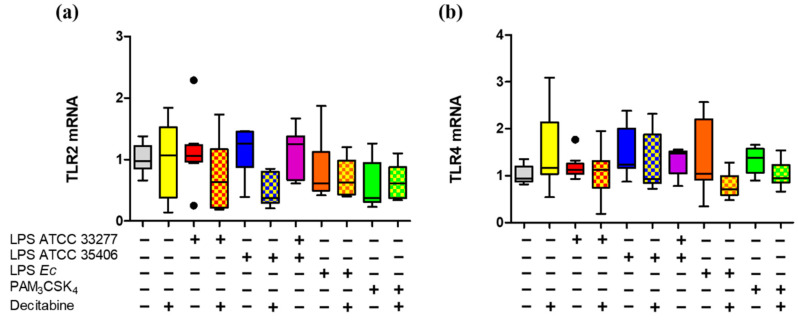
TLR2 and TLR4 gene expression in LPS or lipopeptide-stimulated macrophages in vitro treated with decitabine. Box and whisker plots represent the median and interquartile range of relative gene expression of (**a**) TLR2 and (**b**) TLR4. THP-1 macrophages in vitro were treated with decitabine for 24 h and stimulated for 2 h with LPS or PAM_3_CSK_4_. Relative gene expression was normalized relative to 18S rRNA and expressed as the difference between 2^–ΔΔCt^. Experiments were performed in triplicate. Data were analyzed by the Kruskal-Wallis test. (●) Represents outlier data. (*n* = 3).

**Table 1 ijms-23-10011-t001:** Spearman’s correlation matrix of M1 and M2 surface markers, TLR2, TLR4, and secreted cytokines levels.

	CD64	CD80	CD163	CD206
TNF-α	−0.9 ***	0.7 ***	0.3 ***	0.4 ***
IL-1β	−0.9 ***	0.7 ***	0.3 ***	0.4 ***
IL-6	−0.8 ***	0.7 ***	0.4 ***	0.4 ***
IL-12p70	−0.7 ***	0.8 ***	0.5 ***	0.4 ***
IL-10	−0.8 ***	0.7 ***	0.4 ***	0.4 ***
IL-1RA	−0.5 ***	0.6 ***	0.5 ***	0.3 **
TLR2	0.9 ***	−0.6 ***	−0.3 *	−0.4 ***
TLR4	0.9 ***	−0.6 ***	−0.3 **	−0.4 ***

* *p* < 0.05; ** *p* < 0.005; *** *p* < 0.0005. Values correspond to r (Rho).

## Data Availability

Not applicable.

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
