# Peer review of "Lipopolysaccharide from Porphyromonas gingivalis, but Not from Porphyromonas endodontalis, Induces Macrophage M1 Profile"

_ijms, 2022, doi:10.3390/ijms231710011_

Round 1
Reviewer 1 Report
The authors are investigating the effect of LPS from from Pg and Pe on THP-derived macrophages in order to establish the predominant phenotype of innate immune response – pro-inflammatory M1 or anti-inflammatory/healing-M2 – in endodontic canal infections.
The manuscript is clear and relevant for the field, being presented in a well-structured manner. The experimental design is appropriate to test the hypothesis. The manuscript results are reproducible based on the details provided in the methods section. The methods used are quite complex, from bacterial growth and LPS purification, to THP-derived macrophages cell culture, cytokine secretion, flowcytometric immunophenotype characterization and PCR expression of TLR2 and TLR4 receptors on macrophages.
The figures and schemes/images are appropriate and show the data properly, being easy to understand. The conclusions are consistent with the evidence and arguments presented. The cited references are relevant publications and do not include self-citations.
Just a suggestion for further studies, to test if there is/what type of adaptive immune response can occur in case of infections with these pathogens.
Author Response
Thank you very much for your supportive comments and suggestion.
Yes, in our lab the adaptive immune response induced by periodontal/endodontic pathogens and their respective virulence factors are also part of our labs work.
Reviewer 2 Report
This is very interesting story, but it looks still preliminary to complete author's claim. This reviewer therefore suggest additonal experimental approaches to get more supportive results for completing author's current hypothesis as below.
1. Authors should show the patterns of signaling events (eg., activation signals for NF-kB, STATs and SMAD) under the LPS treatment conditions.
2. Authors also should show the activation patterns of those transcription factors.
3. Cytokine levels at the mRNA levels should be also presented.
4. the cellular activation patterns (eg., phagocytosis and tissue repairing function) of macrophages under P. gingivalis and P. endodontalis LPS treated conditions.
5. Authors should also present M1- or M2-related data with bone marrow-derived macrophages to avoid non-specific changes in cancer conditions.
Author Response
Thank you for your exciting comments and proposals. This study aimed to characterize the polarization profiles -and TLR2/4 expression- of macrophages differentiated from THP-1 cells following challenge with P. gingivalis and/or P. endodontalis LPS from reference strain and clinical isolates on the basis of the following hypothesis: “P. gingivalis and/or P. endodontalis LPS induce a predominant M1 polarization in human macrophages based on their surface markers and cytokine profile”. With that in mind, and as resources and time are limited, we focused on the analysis of the macrophage polarization profiles through a well-validated profiling strategy, based on surface markers and secreted cytokines for each polarization profile [1, 2] also in accordance with the previous characterization performed in human apical lesions [3], rather than the associated pathways and activation patterns.
In line with this, we propose to clafirify this point in the discussion by adding to the 6th paragraph the following:
“During AP, macrophages exhibit a polarization switch towards M1, whereas disease exacerbation during symptomatic AP associates with a reduced M2 differentiation profile based on the reduced surface expression of CD163 along with higher IL-6 and IL-23 levels, which supports a role of macrophage in the severity of apical lesions that can be triggered by endodontic bacterial antigens [32]. It has been previously reported that P. gingivalis LPS induces M1 cytokine secretion in bone marrow-derived macrophages once polarized to M1 or M2 [46], as well as M1 cytokine secretion in THP-1 differentiated to macrophages while being polarized with IFN-γ or IL-4 [47]. However, despite the above, the capacity of P. gingivalis and P. endodontalis LPS to induce macrophage polarization by analyzing their surface markers or cytokine secretion have not yet been described. In the present study, we phenotypically characterized M1 and M2 macrophage subpopulations based on the expression of their cell surface markers CD64 and CD80 for M1, as well as CD163 and CD206 for M2 [32, 48-50].”
In general terms, there are several studies in the literature reporting signaling pathways associated particularly to the stimulus with P. gingivalis LPS. It is also well stablished that TLR2 and TLR4 ligation induced by different ligands, including P. gingivalis LPS, leads to MyD88 activation (and also TRIF activation in the case of TLR4) and subsequent nuclear translocation of the transcription factor NF-κB, triggering cytokine expression [4-10]. Also, the signaling pathways involved in M1/M2 polarization have been characterized [11]. In attention to this, we propose to add in the 3rd paragraph of the Introduction:
“M1 macrophages differentiate from the classical activation pathway triggered mainly by the IFN-γ/Jak–STAT1 pathway under the interplay with TLRs [29,30], playing an important role in fighting bacteria and intracellular pathogens, while M2 macrophages result from exposure to IL-4/IL-13 or IL-10 regulated by STAT6 [31]”.
Regarding mRNA, according to the literature and also to our preliminary experiments, they show peaks levels between 30 minutes and two hours after the LPS stimulus depending on the specific cytokine [4, 12] and after that, the transcription levels return to baseline. Therefore, our 24h model for mRNA quantitation would be inadequate and this would imply the setting of a different stimulation model to perform these analyses. Considering that mRNA levels exclusively provide information on transcribing DNA to mRNA and not on protein production, the secreted cytokines represent a more robust and quantitative outcome which is the one presented here [2].
Although in many cases working with different cells models, THP-1 derived macrophages have been extensively studied for these purposes representing a well-accepted cell model of human macrophages that provides the advantage to eliminate any genetic variability [4, 13-17]. Besides ethical concerns for human donors, periodontitis and apical periodontitis, the oral diseases caused by P. gingivalis and P. endodontalis have a high prevalence in the adult population, which reaches around 90%, which might alter the cell profile of human donors when exposed to bacterial antigens. Animal models, i.e. rodents are much limited to be extrapolated to humans and also have some methodological draw-backs, i.e. they do not share the same surface markers for macrophage polarization.
Finally, our methodologic approach, though limited has been well-validated in the literature and directly supports our conclusions, which are also proposed to be rewritten to:
“In summary, P. gingivalis LPS induces a macrophage polarization profile predominantly towards M1 similarly, although less evident than the generated by E. coli LPS, based on their cell surface markers and secreted cytokines. On the other hand, P. endodontalis LPS did not cause a clear immune response in THP-1 macrophages. Also, P. gingivalis and P. endodontalis LPS stimulation did not induce TLR2 or TLR4 levels changes in THP-1 macrophages. These data suggest that macrophage polarization towards M1 during AP progression might be influenced by the LPS burden of specific pathogens, such as P. gingivalis.”
References
- Ambarus, C. A.; Krausz, S.; van Eijk, M.; Hamann, J.; Radstake, T. R.; Reedquist, K. A.; Tak, P. P.; Baeten, D. L., Systematic validation of specific phenotypic markers for in vitro polarized human macrophages. J Immunol Methods 2012, 375, (1-2), 196-206.
- Ortiz, M. C.; Lefimil, C.; Rodas, P. I.; Vernal, R.; Lopez, M.; Acuna-Castillo, C.; Imarai, M.; Escobar, A., Neisseria gonorrhoeae Modulates Immunity by Polarizing Human Macrophages to a M2 Profile. PLoS One 2015, 10, (6), e0130713.
- Veloso, P.; Fernandez, A.; Terraza-Aguirre, C.; Alvarez, C.; Vernal, R.; Escobar, A.; Hernandez, M., Macrophages skew towards M1 profile through reduced CD163 expression in symptomatic apical periodontitis. Clin Oral Investig 2020, 24, (12), 4571-4581.
- Takashiba, S.; Van Dyke, T. E.; Amar, S.; Murayama, Y.; Soskolne, A. W.; Shapira, L., Differentiation of monocytes to macrophages primes cells for lipopolysaccharide stimulation via accumulation of cytoplasmic nuclear factor kappaB. Infect Immun 1999, 67, (11), 5573-8.
- Azuma, M., Fundamental mechanisms of host immune responses to infection. J Periodontal Res 2006, 41, (5), 361-73.
- Holden, J. A.; Attard, T. J.; Laughton, K. M.; Mansell, A.; O'Brien-Simpson, N. M.; Reynolds, E. C., Porphyromonas gingivalis lipopolysaccharide weakly activates M1 and M2 polarized mouse macrophages but induces inflammatory cytokines. Infect Immun 2014, 82, (10), 4190-203.
- Hirschfeld, M.; Weis, J. J.; Toshchakov, V.; Salkowski, C. A.; Cody, M. J.; Ward, D. C.; Qureshi, N.; Michalek, S. M.; Vogel, S. N., Signaling by toll-like receptor 2 and 4 agonists results in differential gene expression in murine macrophages. Infect Immun 2001, 69, (3), 1477-82.
- Lu, Y. C.; Yeh, W. C.; Ohashi, P. S., LPS/TLR4 signal transduction pathway. Cytokine 2008, 42, (2), 145-151.
- Murray, P. J., Macrophage Polarization. Annu Rev Physiol 2017, 10, (79), 541-566.
- Zhou, Q.; Amar, S., Identification of signaling pathways in macrophage exposed to Porphyromonas gingivalis or to its purified cell wall components. J Immunol 2007, 179, (11), 7777-90.
- Zhou, D.; Huang, C.; Lin, Z.; Zhan, S.; Kong, L.; Fang, C.; Li, J., Macrophage polarization and function with emphasis on the evolving roles of coordinated regulation of cellular signaling pathways. Cell Signal 2014, 26, (2), 192-7.
- Shen, J.; Liu, Y.; Ren, X.; Gao, K.; Li, Y.; Li, S.; Yao, J.; Yang, X., Changes in DNA Methylation and Chromatin Structure of Pro-inflammatory Cytokines Stimulated by LPS in Broiler Peripheral Blood Mononuclear Cells. Poult Sci 2016, 95, (7), 1636-1645.
- Maess, M. B.; Wittig, B.; Cignarella, A.; Lorkowski, S., Reduced PMA enhances the responsiveness of transfected THP-1 macrophages to polarizing stimuli. J Immunol Methods 2013, 402, (1-2), 76-81.
- Lund, M. E.; To, J.; O'Brien, B. A.; Donnelly, S., The choice of phorbol 12-myristate 13-acetate differentiation protocol influences the response of THP-1 macrophages to a pro-inflammatory stimulus. J Immunol Methods 2016, Mar, (430), 64-70.
- Aldo, P. B.; Craveiro, V.; Guller, S.; Mor, G., Effect of culture conditions on the phenotype of THP-1 monocyte cell line. Am J Reprod Immunol 2013, 70, (1), 80-6.
- Daigneault, M.; Preston, J. A.; Marriott, H. M.; Whyte, M. K.; Dockrell, D. H., The identification of markers of macrophage differentiation in PMA-stimulated THP-1 cells and monocyte-derived macrophages. PLoS One 2010, 5, (1), e8668.
- Zarember, K. A.; Godowski, P. J., Tissue expression of human Toll-like receptors and differential regulation of Toll-like receptor mRNAs in leukocytes in response to microbes, their products, and cytokines. J Immunol 2002, 168, (2), 554-61.
Round 2
Reviewer 2 Report
OK, authors have fully addressed. So, it is now acceptable.